# Perception of Thermal Comfort during Skin Cooling and Heating

**DOI:** 10.3390/life11070681

**Published:** 2021-07-12

**Authors:** Igor B. Mekjavic, Daniel Yogev, Urša Ciuha

**Affiliations:** Department of Automation, Biocybernetics and Robotics, Jozef Stefan Institute, SI-1000 Ljubljana, Slovenia; yogevdan@gmail.com (D.Y.); ursa.ciuha@ijs.si (U.C.)

**Keywords:** thermal comfort, temperature perception, behavioral temperature regulation, thermoregulatory model

## Abstract

Due to the static and dynamic activity of the skin temperature sensors, the cutaneous thermal afferent information is dependent on the rate and direction of the temperature change, which would suggest different perceptions of temperature and of thermal comfort during skin heating and cooling. This hypothesis was tested in the present study. Subjects (N = 12; 6 females and 6 males) donned a water-perfused suit (WPS) in which the temperature was varied in a saw-tooth manner in the range from 27 to 42 °C. The rate of change of temperature of the water perfusing the suit (T_WPS_) was 1.2 °C min^−1^ during both the heating and cooling phases. The trial was repeated thrice, with subjects reporting their perception of the temperature and thermal comfort at each 3 °C change in T_WPS_. In addition, subjects were instructed to report when they perceived T_WPS_ uncomfortably cool and warm during cooling and heating, respectively. Subjects reproducibly identified the boundaries of their Thermal Comfort Zone (TCZ), defined as the lower (T_low_) and upper (T_high_) temperatures at which subjects reported slight thermal discomfort. During the heating phase, T_low_ and T_high_ were 30.0 ± 1.5 °C and 35.1 ± 2.9 °C, respectively. During the cooling phase, the boundary temperatures of T_low_ and T_high_ were 35.4 ± 1.9 °C and 38.7 ± 2.3 °C, respectively. The direction of the change in the cutaneous temperature stimulus affects the boundaries of the TCZ, such that they are higher during cooling and lower during heating. These findings are explained on the basis of the neurophysiology of thermal perception. From an applied perspective, the most important observation of the present study was the strong correlation between the perception of thermal comfort and the behavioral regulation of thermal comfort. Although it is not surprising that the action of regulating thermal comfort is aligned with its perception, this link has not been proven for humans in previous studies. The results therefore provide a sound basis to consider ratings of thermal comfort as reflecting behavioral actions to achieve the sensation of thermal neutrality.

## 1. Introduction

Human normothermia is maintained by activation of both autonomic and behavioral thermoregulatory responses. Behavioral responses are the most efficient of thermoregulatory responses, as relatively simple actions can alter the microclimate surrounding the body and prevent the need to activate the more metabolically costly autonomic responses. Such responses in humans may involve a wide variety of actions including moving to a different thermal ambiance, changing posture, changing clothing and/or activating climatic control systems (e.g., air-conditioning in dwellings, heating/cooling systems under protective clothing). 

Behavioral thermoregulatory responses are initiated once the surrounding environment is no longer perceived as thermally comfortable or pleasurable [1,2,3]. Assessment of behavioral thermoregulation in humans relies mainly on the evaluation of changes in the perception of thermal comfort. Thermal comfort occurs when an individual expresses indifference to the thermal environment [4]. This ability of humans to report how comfortable/uncomfortable they perceive their thermal environment has been utilized to develop qualitative assessment methods in which the intensity of thermal perception (thermal comfort or temperature perception) is expressed on a rating scale [5,6,7,8,9]. Qualitative data of this kind can be very useful for estimating the conditions where most humans would feel comfortable in an everyday environment (e.g., in buildings or automobiles). However, for evaluating the effects of extreme environments (e.g., underwater, space) or a specific parameter within it (e.g., hyperbaric pressure, nitrogen narcosis, etc.) on the risk of thermal instability, the use of qualitative tools can be considered inadequate. It is well known that the results of scale votes are influenced by psychological factors including leading questions, question specificity, language, embarrassment, ego, etc. [10].

Thermal (dis)comfort is considered to be the driving force for the initiation of behavioral thermoregulatory responses [3]. However, if thermal perception is altered (e.g., by a nonthermal factor) and behavioral responses are not activated appropriately, maintenance of normothermia might be jeopardized [11]. Altered perception of thermal comfort has been documented with various nonthermal factors including gender [12,13,14,15,16], aging [17,18,19,20], inert gas narcosis [21,22,23], hypoglycemia [24], hypoxia [25,26], and sleep deprivation [27]. With few exceptions [19,25,26,28,29,30], these studies evaluated thermal comfort with visual analog scales, and speculated on the behavioral actions that would be initiated by the reported subjective ratings of thermal comfort. Studies introducing interventions to evaluate the effect of specific nonthermal factors, interpreted the nonthermal factor-induced changes in subjective ratings of comfort as evidence that behavioral modifications would be affected in a similar manner. The effect of nonthermal factors on behavioral thermoregulatory actions has been confirmed in animal studies [3,31,32,33,34,35]; however, data in humans is still lacking. 

Various experimental approaches have been suggested for assessing thermal comfort regulation. For example, adjustments in the level of clothing (dressing behavior) were quantified to assess the influence of factors such as age [36], menstrual cycle [37], and light intensity [38]. However, even if such behavioral measures correlate well to changes in thermal conditions, the cause for such behavior might be influenced by modesty, acceptability and design [10]. Other, direct measurements of thermoregulatory behavior in controlled conditions have been described using operant conditioning methods. In operant conditioning methods, the subject is given voluntary control over some aspect of the surrounding thermal environment. Although operant conditioning methods have been largely used in animal studies [3,31,32,33,34,35,39] several studies, however, found them applicable also for human subjects. To study the effect of age on behavioral thermoregulation, Collins et al. [19], for example, allowed young and old subjects to control room temperature to their preferred level using a remote controller. In the experiments of Webb et al. [40] and Hexamer and Werner [41] subjects could control a water cooling garment manually, while their objective physiological state was simultaneously analyzed. During exercise [41] the subjects were capable of maintaining thermal comfort, but manual control of the water temperature based on subjective perception of thermal sensation was not adequate to prevent displacements of core temperature, suggesting that manual/subjective control is not optimal for maintaining thermal balance in such conditions. 

Common to most studies investigating subjective ratings of the perception of temperature and thermal comfort is the assumption that such perceptions are dependent on the combination of skin and core temperatures, disregarding the possibility that the direction of the change in temperature may also exert an effect on these perceptions. Animal studies [42,43] have demonstrated that the response of a temperature sensor to a step change in temperature is a dynamic overshoot or undershoot in activity, which subsides to a steady state level of tonic activity, corresponding to the adaptation temperature. Cooling will increase the dynamic activity of the cold and decrease dynamic activity of the warm sensors. Conversely, warming will increase and decrease the dynamic activity of the warm and cold sensors, respectively. As the central perception of temperature is dependent on the thermoafferent information from the peripheral sensors, which in turn is dependent on the magnitude of the thermal stimulus, its rate of change and direction (i.e., heating or cooling), the present study tested the hypothesis that the direction of the temperature change will influence the perception of temperature and thermal comfort. It further tested the hypothesis that the perception of thermal discomfort is correlated with behavioral actions to counteract the discomforting thermal stimulus. The test of this latter hypothesis is fundamental to all studies using visual analog scales of thermal discomfort, as an index of behavioral temperature regulation. Namely, should behavioral modifications to minimize thermal discomfort not match the ratings reported by subjects, then this would invalidate visual analog scales as an index reflecting behavioral temperature regulation.

## 2. Materials and Methods

The experimental protocol was approved by the National Medical Ethics Committee of the Slovene Ministry of Health, and conformed to the Declaration of Helsinki. Subjects gave their informed consent to participate in the study. A total of 12 healthy subjects (6 males, 6 females) participated in these trials conducted at the Jozef Stefan Institute. Their mean ± SD physical characteristics were: age = 25.2 ± 3.9 years (males: 25.8 ± 5.4 years; females: 25.5 ± 1.4 years), height = 176.4 ± 9.8 cm (males: 182.8 ± 7.5 cm; females: 170.6 ± 8.1 cm), weight = 68 ± 14.6 kg (males: 78.7 ± 11.6 kg; females: 59.5 ± 7.9 kg).

### 2.1. Experimental Protocol

Subjects participated in two experimental trials. The order of the two trials was randomized. In both, the subjects donned a water-perfused suit (WPS) with a control unit designed to allow either the subject, or the experimenter, control over the temperature of the water perfusing the WPS (Figure 1). Specifically, by depressing a control switch, the subject (or experimenter) could change the direction of the temperature of the water perfusing the WPS. The temperature control unit had no steady-state position, thus, the temperature of the WPS alternated between a cooling and heating mode according to the subject’s, or experimenter’s control. Using this experimental arrangement, two separate trials were conducted. In both, the experimental session commenced with a 30 min acclimation to room conditions (ambient temperature, Ta = 25 °C; relative humidity, RH = 30%) and familiarization with the equipment and study protocol. Then, subjects were instrumented with skin sensors, donned a WPS and assumed a supine position on a gurney. Male subjects were dressed in shorts, and females in shorts and a bikini top.

#### 2.1.1. Perception of Thermal Comfort

The temperature of the water perfusing the WPS (T_wps_) was initially regulated at a baseline temperature of 27 °C for 10 min. Thereafter, subjects were exposed to a heating and cooling protocol that was repeated thrice in sequence, during which T_wps_ varied from 27 to 42 °C and back, at a rate of 1.2 °C.min^−1^. Prior to the onset of the trials, subjects were informed of the nature of the thermal stimuli that would be administered, but were not given any verbal feedback during the trial.

At each 3 °C change in temperature subjects were requested to rate their thermal perception on a 7-point scale (−3: very cold; −2: cold; −1: slightly cold; 0: neutral; +1: slightly warm; +2: warm; +3: very warm) and thermal discomfort on a 4-point scale (0: comfortable; 1: slightly uncomfortable; 2: uncomfortable; 3: very uncomfortable). Furthermore, they were asked to report when they perceived the temperature change from a comfortable to an uncomfortable level and vice versa, thus indicating the boundaries (T_low_ and T_high_) of their thermal comfort zone (TCZ) during heating and cooling. During the trials, subjects were reminded of these instructions at regular intervals.

#### 2.1.2. Regulation of Thermal Comfort

As in the previous trials (perception of thermal comfort), T_wps_ was regulated by the control unit to vary between 27 °C and 42 °C, at a rate of 1.2 °C.min^−1^. In contrast to the perception of thermal comfort trials, in these trials subjects were instructed that they could initiate a change in the direction of the temperature of the water perfusing the WPS once it became either uncomfortably warm during heating, or uncomfortably cool during cooling, by depressing a button on a manual control switch. Subjects were instructed to maintain T_wps_ within a preferred range for a total of one hour. The subjects were reminded of these instructions at regular intervals. These trials provided a characteristic saw-tooth T_wps_ pattern indicating the lower (T_low_) and upper (T_high_) temperature boundaries, and thus also the range of the thermal comfort zone (TCZ) during heating and cooling.

In all trials, subjects were naïve regarding the absolute temperature of the water perfusing the WPS. They could only assess their thermal (dis)comfort (self-reporting, ranking on a scale, depressing a switch) according to their subjective assessment of the temperature change.

### 2.2. Instrumentation

#### 2.2.1. Water Perfused Suit (WPS)

The WPS comprised five sections: one covering the head and upper back, one for each leg, one surrounding the torso, and one for both arms and lower back. The WPS did not cover the hands, feet, neck and face. Water to all five sections of the WPS was delivered via a common manifold and comprised identical lengths (25 m) of small diameter (inner diameter = 4 mm, outer diameter = 5 mm) polyvinylchloride (PVC) tubing, which were woven into the meshed lining of the suit. This ensured equal flow of water in all five segments of the suit of approximately 3.5 L.min^−1^. The total volume of water contained by the tubes of the suit was approximately 1.25 L. The WPS was designed to fit various sizes by adjusting the Velcro fasteners.

#### 2.2.2. Temperature Control of WPS

The WPS manifold was connected to two 5 L warm and cold water reservoirs, respectively. The water from these reservoirs was pumped by two pumps (Barwig, Conrad Electronics, Hirschau, Germany) to a 1 L mixing container before being circulated by a third pump to the WPS. Dedicated computer software enabled the control of the T_wps_. The controller regulated the rate of change of the T_wps_ when the heating and cooling modes were activated. The controller could also be set to automatically maintain a saw-tooth oscillation in the T_wps_. T_wps_ was measured using T-type thermocouples positioned at the inlet (T_in_) and the outlet (T_out_) of the suit. The average between T_in_ and T_out_ was considered as T_wps_. A schematic representation of the experimental arrangement is presented in Figure 1.

#### 2.2.3. Ambient Conditions

During the trials, ambient temperature (Ta, °C), pressure (Pb, mmHg) and relative humidity (RH, %) were measured with a portable weather station (BAR 938 HG OS, Huger, Frankfurt, Germany).

#### 2.2.4. Skin and Core Temperature

Skin temperature (Tsk, °C) was measured using thermocouples (Concept Engineering, Old Saybrook, CT, USA), which were attached at eight sites (toe, calf, thigh, abdomen, chest, fingertip, forearm, and arm). The thermocouples were attached to the skin on the right side of the body using a thin, breathable transparent adhesive film (3M™ Tegaderm™). Skin temperature data were collected at 1 second intervals on a data logger (Almemo 5990-2, Ahlborn, Holzkirchen, Germany). Core temperature (T_c_, °C) was estimated with an infrared tympanic temperature (ThermoScan IRT 3020, Braun, Kronberg, Germany). We have previously observed good correlations of measurements of tympanic temperature measured with an infrared tympanometer with rectal (0.95 ± 0.03) and esophageal (0.96 ± 0.02) temperatures [44]. Tympanic temperature was monitored only to ensure that there were no changes in core temperature during the course of the trials, as anticipated on the basis of our earlier studies [28,29,30].

### 2.3. Calculations and Data Analysis

The reproducibility of identifying the boundaries of the TCZ was evaluated using a 1 × 3 ((males + females) × (Trial 1/Trial 2/Trial 3)) two-way analysis of variance (ANOVA) with repeated measures on the lower limit (T_low_), the upper limit (T_high_) and the width of the TCZ between heating and cooling phases in trials 1–3. Subjective ratings are non-normally distributed, and thus the differences between the medians of the ratings of thermal perception and thermal comfort were analyzed using the non-parametric Kruskal–Wallis analysis of ranks. A paired t-test was used to compare between T_wps_ at the moment discomfort was reported and the moment a behavioral response was initiated. T_c_ was measured before and immediately after each trial. Data were expressed as mean ± SD and the limit of statistical significance was set to 0.05. Based on the results of our previous studies, we determined that for a required power of a statistical test of 0.8, at least 12 participants would be required to ensure that the upper and lower threshold temperatures of the TCZ could be defined as significantly different at a 0.05 significance level using a two-tailed test. Sex-related differences regarding thermal comfort were not explored due to the small sample size involved.

## 3. Results

All subjects detected the change in the temperature of the water perfusing the suit during Trial 1 (Perception of thermal comfort), and were capable of regulating the thermal comfort, as requested in Trial 2 (Thermal comfort regulation). During these trials, T_c_ remained unchanged. 

### 3.1. Characteristics of Temperature Changes in the WPS

The temperature control system changed T_wps_ at a rate of 1.2 °C min^−1^ during both the heating and cooling phases. The largest difference between T_in_ and T_out_ of the WPS (approximately 1−2 °C) was during the shift from heating to cooling, and vice versa.

### 3.2. Thermal Comfort Vote (TCV) and Temperature Perception Vote (TPV) during Prescribed Conditions

Subjects were requested to rate their perception of the temperature and thermal (dis)comfort at each 3 °C change in the T_wps_ during heating and cooling in the range 27 to 42 °C. There were no significant differences between the median votes obtained from the three trials over the given range of temperatures. During the heating process, the lowest ratings of discomfort were reported at T_wps_ = 30 °C and 33 °C (Figure 2A). These values were perceived as “neutral” and “slightly warm” (Figure 2B), respectively. Similarly, during cooling subjects felt most comfortable at T_wps_ = 36 and 39 °C (Figure 3A), which also corresponded with perceptions of “neutral” and “slightly warm” (Figure 3B), respectively. 

In addition to providing ratings of thermal comfort and temperature perception, subjects were also requested to indicate transitions between thermal comfort and discomfort. In the three trials, subjects reported the boundaries of their TCZ reproducibly. There were no significant differences (*p* > 0.05) between T_low_, T_high_ and the width of TCZ as derived in three repeat trials, confirming the repeatability of the test. The boundaries of TCZ during warming were 30.0 ± 1.5 °C (T_low_) and 35.1 ± 2.9 °C (T_high_), and during cooling 35.4 ± 1.9 °C (T_low_) and 38.7 ± 2.3 °C (T_high_). The coefficient of variation, reflecting the measure of accuracy of the T_wps_ eliciting discomfort, was 2.3% for cold discomfort (T_low_ during cooling), and 5.5% for warm discomfort (T_high_ during heating).

### 3.3. Thermal Comfort Regulation during Individual Control

In the thermal comfort regulation trial subjects were not asked to provide any ratings. Their only task was to regulate T_wps_, by altering the direction of T_wps_ when it was perceived as uncomfortable. In this manner, they defined the boundaries of their TCZ. The threshold T_wps_ at which subjects initiated a behavioral response to counteract thermal discomfort was determined by calculating the average T_wps_ at which subjects activated the control switch during cooling and warming. The average T_wps_ at which subjects interrupted with heating (T_high_) was 34.7 ± 1.0 and the T_wps_ at which they responded to cooling (T_low_) was 34.1 ± 0.8.

### 3.4. Correlation between Perceptual Change and Behavioral Response

There was no significant difference between the T_wps_ at which discomfort was reported (Thermal comfort perception trial) and the T_wps_ at which behavioral responses were initiated (Thermal comfort regulation trial). Figure 4A shows the relation between the T_wps_ subjects reported as perceiving uncomfortably cool during cooling, and the T_wps_ at which they altered the direction of T_wps_ to heating. Similarly, Figure 4B shows the relation between the T_wps_ subjects reported as perceiving uncomfortably warm during heating, and the T_wps_ at which they altered the direction of T_wps_ to cooling. There was a strong correlation between these perceptual and behavioral T_wps_ thresholds. The Pearson correlation coefficient was 0.73 during heating (Figure 4A) and 0.84 during cooling (Figure 4B).

## 4. Discussion

The principal finding of the present study is that subjects could reproducibly define the boundaries of their TCZ, and that these boundaries were higher during the application of a cooling stimulus compared to those observed during the application of a heating stimulus. The heating and cooling stimuli had the same rate of change, which leads us to conclude that the difference in TCZ between the cooling and heating stimuli is related to the direction of the temperature change. Based on existing neurophysiological evidence regarding the functional characteristics of the cutaneous cold and warm sensors, we propose a hypothesis that would explain this novel finding.

From an applied perspective, the most important, but seemingly inconsequential observation of the present study, is the strong correlation between the perception of thermal comfort and the regulation of thermal comfort. Although not surprising that the action of regulating thermal comfort is aligned with its perception, this link has not been proven for humans. The results therefore provide a sound basis for considering ratings of thermal comfort as reflecting behavioral actions to achieve the sensation of thermal neutrality. This interpretation should be limited to the conditions prevailing in the present study, which did not impose any changes in core temperature, but only applied a saw-tooth thermal stimulus to the skin.

### 4.1. Visual Analog Scales as an Index of Behavioral Temperature Regulation

Studies investigating behavioral temperature regulation in humans rely on VAS to provide ratings of perceived thermal comfort. It is then assumed that these ratings reflect behavioral actions that would be initiated by the individual to maintain, or re-establish, thermal equilibrium. The present results indicate that the direction of the change of the thermal stimulus significantly affects the perception of thermal comfort, whereby neutral temperature was reported at 30 °C during heating and at 36 °C during cooling. Furthermore, the ratings of thermal comfort were also affected, such that the TCZ was between approximately 30 and 35 °C during heating, and between 38 and 35 °C during cooling. 

### 4.2. Neurophysiological Correlate of Thermal Comfort—A Hypothesis

The effect of the direction of the temperature change on temperature perception and thermal comfort may be explained on the basis of the neurophysiology of thermoreception. Cortical integration of thermoafferent information elicits a conscious assessment of whether the thermal status of the skin is pleasant or unpleasant. We have previously reported that the range of temperatures considered thermally comfortable is centered around 35 °C, which coincides with the region of overlap of the static firing characteristics of the warm and cold sensors [45], as shown in Figure 5. According to Bazett [46] and Vendrik [47], this point of overlapping activity of the cold and warm sensors could be considered as a peripheral “reference” temperature, deviations from which would activate behavioral responses. Teleologically such an arrangement would make sense, as decreases and increases in temperature of the skin would be associated with increased firing rate of the cold and warm receptors, respectively. Two perception thresholds are postulated: perception of a thermal stimulus, and perception of (dis)comfort. The stimulus perception threshold corresponds to the minimal temperature change that can be perceived by an individual. The temperature at the perception threshold may not necessarily be perceived as uncomfortable. According to the hypothesis presented in Figure 5, the thresholds for warm and cool discomfort may require a greater thermal stimulus, i.e., they would occur as a consequence of greater temperature receptor activity. It may be that the perception of thermal discomfort formulated in the sensory cortex is related to a threshold of sensor activity, or rather to the perception of this threshold. Assuming that a same level of activity of the cutaneous cold and warm sensors is associated with a threshold perception of cold and warm thermal (dis)comfort, respectively, then this range of temperatures could provide a neurophysiological correlate of the TCZ. Assuming that the static and dynamic characteristic of the cutaneous thermoreceptors [42,43] are similar in different regions of the skin, then this would imply that the neurophysiological correlate of the region of thermal comfort (i.e., region of equivalent firing rates) would also be the same. This line of reasoning is certainly supported by the results of the present study. As yet, the threshold level of sensor activity required for the initiation of the perception of thermal (dis)comfort remains unresolved. As suggested in Figure 5, it is unlikely that the perception of warm and cold (dis)comfort is centered at one temperature, but that separate threshold temperatures exist for cold and warm (dis)comfort. These threshold temperatures describe the limits of the TCZ. The limits, and thus the range, of the TCZ is most likely affected by the rate of change of skin temperature, as this influences the dynamic activity of the sensors. 

To understand the shift in the boundaries of TCZ it is necessary to appreciate that the thermoreceptors have static and dynamic responses, which are centrally integrated. As explained by Zotterman [48], the effect of cooling would be an additional increase in the firing rate of the cold and an additional decrease in the firing rate of the warm sensors (Figure 6, left panel), whereas warming would cause an additional increase in firing rate of the warm sensors and an additional decrease in the activity of the cold sensors (Figure 6, right panel). As shown in Figure 6 (left panel), cooling would thus result in a shift of the region of equal and overlapping activity of the cold and warm sensor to a higher temperature, whereas during heating (Figure 6, right panel) this region of equal and overlapping activity would be shifted towards lower temperatures. This would explain, to a degree, the differences regarding TCZ in the present study.

According to the theory of reciprocal cross inhibition (RCI) proposed by Bligh [49], the thermal afferent information from cold and warm sensors, provides the neural drive for heat production and heat loss, respectively. The basic tenet of this theory is that the excitatory drives from the cold and warm sensors not only stimulate heat production and heat loss effectors, respectively, but via inhibitory synapses the thermoafferent drive from one sensor inhibits the thermoafferent drive from the other sensor. In the context of the neural model presented, this would suggest that during cooling the increased activity of the cold sensors would also provide an inhibitory stimulus to the thermoafferent pathway from the warm sensors. Similarly, heating would increase the warm sensory activity, and concomitantly inhibit the thermoafferent pathway from cold sensors. RCI would therefore act to further reduce the activity of the warm sensors on cooling and the cold sensors on warming, resulting in a more pronounce shift of the overlapping region of equal activity of the sensors.

### 4.3. Validity of Visual Analog Scales for Thermal Comfort

The good correlation between the threshold T_wps_ at which subjects reported perceiving the transition from thermal comfort to discomfort during heating and cooling and the upper and lower boundaries of the TCZ determined with the self-regulated T_wps_ provides validation of the visual analog scales used to determine thermal (dis)comfort. Namely, the results indicate that at temperatures perceived as thermally uncomfortable subjects initiated a behavioral action to counteract the perceived thermal discomfort.

### 4.4. Anticipatory Response

The most interesting finding of the present study is the apparent hysteresis in the perception of thermal comfort as a function of T_wps_. As illustrated in Figure 7, the transition from the perception of thermal discomfort to comfort during cooling (Figure 7, left panel) is perceived at a higher temperature than during heating. Also, T_wps_ considered thermoneutral is higher during cooling than during heating (Figure 7, right panel) Although the proposed hypothesis provides an explanation for the likely neural mechanism, it does not address the benefit of such neurophysiological arrangement. We propose that the observed response may represent a feed-forward mechanism [50], by which anticipatory behavioral actions are activated before they are required from a thermal balance perspective. Specifically, during cooling from 42 °C the thermal comfort equilibrium point is shifted towards high skin temperatures, due to the augmentation and attenuation of the cold and warm receptors, respectively. As a consequence, behavioral actions against cold stress will be initiated at a skin temperature that would normally be perceived as lightly warm or thermally comfortable. In contrast, during heating from 27 °C, thermal discomfort will be perceived at a lower skin temperature, initiating behavioral actions in anticipation of the potential heat stress.

### 4.5. Practical Implications

The results of the present study, summarized in Figure 7, could provide an explanation regarding the thermal comfort perceived during sweating. Namely, the evaporation of sweat from a warm skin surface provides a cooling effect of the skin [51], but not necessarily of a magnitude, that would decrease skin temperature to levels normally considered thermally neutral and comfortable. We speculate that, despite an elevated skin temperature, the cooling provided by evaporation is perceived pleasant as a consequence of the TCZ being shifted to higher skin temperatures, as illustrated in Figure 7. The basis for this shift in the TCZ towards high temperatures, is provided by the neural model of thermal comfort presented above (Figure 6), suggesting that the dynamic characteristics of the cutaneous temperature sensors could explain the cooling- and heating-induced shifts in the TCZ.

Maintenance of thermal comfort is of paramount importance to maintain well-being, particularly during work. Rather than try and implement costly air conditioning systems that strive to maintain low air temperatures, it may be possible to establish thermally comfortable working environments with fluctuating air temperatures. The present results would suggest that, at least during the cooling periods, thermal comfort might be established at high air temperatures.

### 4.6. Limitations

The principal limitation of the present study is the manner in which skin temperature was measured. Namely, sensors were placed on the skin at sites recommended by Ramanathan et al. [52], and the placement did not coincide with the position of the water-perfused tubes in the suit. The responses of the subjects were therefore analyzed on the basis of the temperature profile of the water perfusing the suit. This issue of whether, and how, skin temperature reflects the temperature of such cooling/heating garments is not only an issue of this, but also of all studies concerned with assessing the effect of cooling/heating garments on skin temperature.

The apparent incongruence of the thermal stimuli and skin temperature responses, and its potential effect on the thermal perception results has been allayed by the results of our recent study, which has examined the same perceptual responses during variations in air temperature [53].

The saw-tooth nature of the thermal stimulus presented to the subjects with the WPS was not of a magnitude that would be anticipated to affect core temperature. Thus, the use of infrared tympanometry as an index of T_c_ may certainly be considered a limitation of the study, but the measurement was used only to verify that core temperature was stable throughout the trials. Infrared tympanometry was used to ensure no relative change in core temperature, and was not used as a measure of the absolute temperature of the core region.

Finally, the current study included male and female subjects, but statistical comparisons of thermal perception and thermal comfort were not conducted due to the small sample size. This is an important issue and should be further explored with studies designed specifically to investigate sex-related differences in the perception of thermal comfort.

## 5. Conclusions

On the basis of the physiological evidence of a hysteresis in the perception of thermal comfort during cooling and warming we propose a thermal comfort model based on the static and dynamic firing response characteristics of the cutaneous cold and warm sensors. This neural model of thermal perception suggests a feedforward mechanism that would provide anticipatory behavioral actions counteracting cold strain during cooling, and heat strain during warming.

We previously demonstrated that the prediction of shivering thermogenesis [54] was improved with a model incorporating both the static and dynamic responses of the central and cutaneous temperature receptors [47]. The current study proposes that consideration of the dynamic response of the cutaneous cold and warm sensors may provide an improved prediction of thermal comfort. Further studies are required to also document the manner in which nonthermal factors [55,56] that impact on the thermoregulatory neuraxis modify the perceptions of thermal comfort and temperature sensation. 

Two such important factors are gender and aging. With regards to the former, sex hormones have been demonstrated to impact on autonomic [57] and behavioral [58] temperature regulation, specifically on cutaneous thermal sensitivity and the perception of thermal comfort [58]. 

Global warming has caused an increase in the occurrence of devastating summer heat waves. These periods of alarming elevations in ambient temperature affect all aspects of our lives and can have fatal consequences for particular high risk groups. Strategies and technologies need to be developed in the face of this crisis to maintain thermal comfort and thus prevent disruption of our livelihood and lives. 

## Figures and Tables

**Figure 1 life-11-00681-f001:**
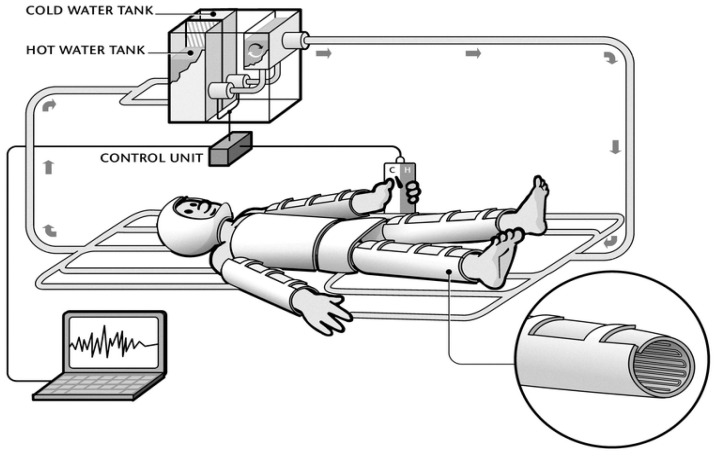
A schematic representation of the experimental arrangement used to evaluate comfort regulation as a measure of behavioral thermoregulation. By depressing a button, the subject or experimentor could change the direction of the temperature perfusing the WPS. In the Regulation of Thermal Comfort trial (see text), subjects were instructed to depress the control switch, thus altering the direction of the temperature change, when they perceived the temperature becoming slightly uncomfortable.

**Figure 2 life-11-00681-f002:**
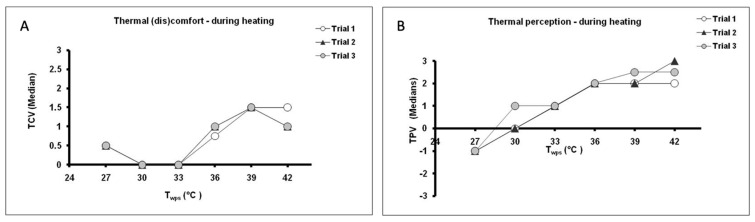
Subjective ratings (median values) of (**A**) thermal (dis)comfort (thermal comfort vote, TCV) and (**B**) thermal perception (temperature perception vote, TPV) as a function of the temperature of the water perfusing the suit (T_wps_) during the heating phase in trials 1–3. At each 3 °C change in temperature subjects were requested to rate their thermal perception (TPV) on a 7-point scale (−3: very cold; −2: cold; −1: slightly cold; 0: neutral; +1: slightly warm; +2: warm; +3: very warm) and thermal comfort (TCV) on a 4-point scale (0: comfortable; 1: slightly uncomfortable; 2: uncomfortable; 3: very uncomfortable).

**Figure 3 life-11-00681-f003:**
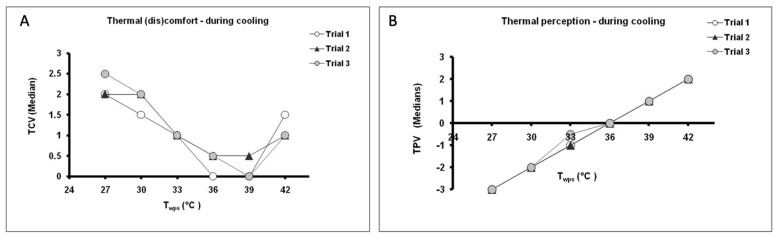
Subjective ratings (median values) of (**A**) thermal (dis)comfort (thermal comfort vote, TCV) and (**B**) thermal perception (temperature perception vote, TPV) as a function of the temperature of the water perfusing the suit (T_wps_) during the cooling phase in trials 1-3. At each 3 °C change in temperature subjects were requested to rate their thermal perception (TPV) on a 7-point scale (−3: very cold; −2: cold; −1: slightly cold; 0: neutral; +1: slightly warm; +2: warm; +3: very warm) and thermal comfort (TCV) on a 4-point scale (0: comfortable; 1: slightly uncomfortable; 2: uncomfortable; 3: very uncomfortable).

**Figure 4 life-11-00681-f004:**
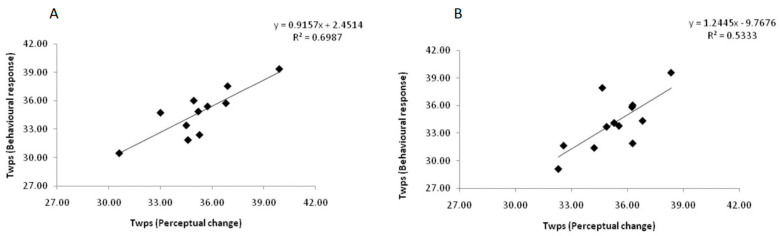
Correlation between the threshold T_wps_ during warming (**A**) and cooling (**B**) at which discomfort was reported (Thermal comfort perception trial) and the threshold T_wps_ at which a behavioral response to counteract discomfort was initiated (Thermal comfort regulation trial).

**Figure 5 life-11-00681-f005:**
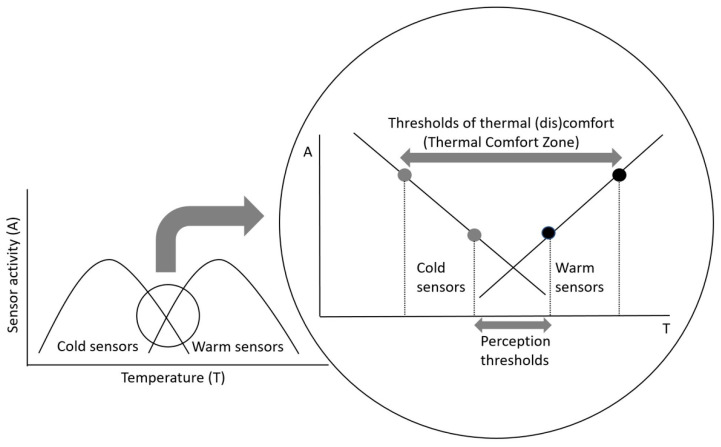
The activity of the cold and warm sensors in response to a steady state temperature stimulus. The region of overlapping, and thus equal activity of the sensors is likely the temperature of thermoneutrality. Decreasing and increasing temperature from this point of overlapping activity initiates increased activity of the sensors, which must reach a certain perception threshold for it to be perceived centrally. The grey and black dots represent the threshold for the perception of cold and warm (dis)comfort, and thus form the limits of the thermal comfort zone.

**Figure 6 life-11-00681-f006:**
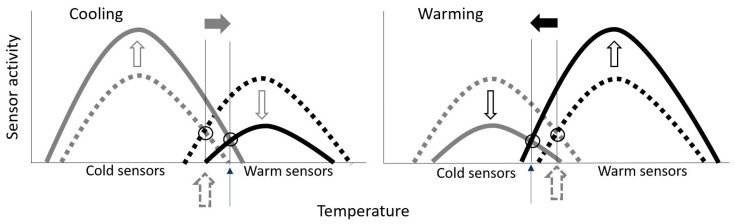
Cooling of the cutaneous sensors will initiate a dynamic response, which will transiently cause a decrease in the warm and increase in the cold sensor activity (**left panel**) compared to the activity observed during steady state temperature (dashed line). In contrast, warming (**right panel**) will cause an increase in the warm sensor activity and a decrease in the cold sensor activity from steady state values. As a consequence of these effects of dynamic changes in temperature on the sensor responses, the thermoneutral temperature would be shifted to a high temperature during cooling (**left panel**) and to a lower temperature during warming (**right panel**).

**Figure 7 life-11-00681-f007:**
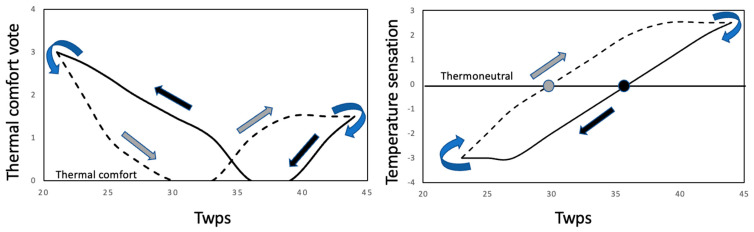
Illustration of the hypothesized hysteresis responses of the perception of thermal comfort (**left panel**) and temperature sensation (**right panel**) to heating (grey arrows) and cooling (black arrows) stimuli provided by the water perfused suit (WPS). The perception of thermal comfort (**left panel**, rating = 0) and thermoneutrality (**right panel**, rating = 0) occurred at higher temperatures of the water perfusing the WPS (T_wps_) during cooling (black lines and arrows) compared to heating (dashed lines and grey arrows).

## Data Availability

The data presented in this study are available on request from the corresponding author.

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
