# Peer review of "Perception of Thermal Comfort during Skin Cooling and Heating"

_life, 2021, doi:10.3390/life11070681_

Round 1

Reviewer 1 Report

Reviewer’s comments (submitted paper ID:life-1252446)

This study investigated whether the feelings of thermal comfort and sensation during exposure to alternating hot and cold environments (that they could not influence; first set of experiments) coincides with the preference of the participants (in another set of experiments) to reverse the direction of the temperature change in the water perfused suits when participants thought that it was too hot or cool. A strong correlation between the upper and lower boundaries of the thermal comfort zone observed in both experiments indicated that human participants align the action of regulating thermal comfort with its perception. Furthermore, the authors demonstrated that the direction of change in the water temperature flowing inside a water-perfused suit is another determinant factor of thermal comfort and sensation added to the absolute temperature of the water. The authors should be commended for completing the experiments of an insightful study featuring an elegant experimental design. The reviewer’s specific comments are attached below.

L63-64: Sleep deprivation should probably be mentioned

L121 and L128: ‘two experimental trails’ and ‘two separate studies were conducted’. These statements are confusing, though I assume authors intend to mean the same.

L143: ‘heating and cooling protocol was repeated thrice’. Details are needed. How was it performed? Was the order ‘Going upwards, downwards and upwards’ or any other possible combination? Was it in three separate times?

L144: How was the rate of change in the wps temperature as well as the upper and lower boundary for the water temperature selected? To my understanding, it seems that a fastest rate of wps water temperature increase during the heating phase would shift the thresholds of sensation to a lower level and the opposite would stand for the cooling phase.

L164: In addition to the absolute temperature of the water flowing in and out of the suit, were participants further naïve regarding the direction of change (at any given moment) in the WPS temperature, during the heating and cooling phases?

L201-203: The present reviewer and the majority of the literature are not convinced that the tympanic temperature is a valid index for core temperature. This has to be mentioned, as a limitation of the study. If the reviewer is not mistaken, tympanic temperature data were not presented in the results section. The authors are requested to modify the section accordingly.

L205: Data should be checked whether are normally distributed and respective results should be reported.

L217-219: This statement should be modified. The argument would rather be that the sex-related differences regarding thermal comfort and sensation were not explored in the current manuscript due to the small sample size involved. The mix of gender with such small size, in a topic were sex effect could be present (thermal sensation) is considered a weak point of the study and should be mentioned in the discussion section.

L260-264: This part of the text seems better suited for the methods rather than the results section.

L305-309: Results repeated in the discussion section. Core temperature values should be provided in the results section

L349: There are no blue and red dots as the picture is given in gray scale; the caption should be modified accordingly.

L396-404: Reviewer agrees with the authors that the hysteresis in the perception of thermal comfort as a function of the skin temperature acts as a feed-forward mechanism meant to activate behavioural actions before the metabolically costly autonomic regulation comes into play. Werner’s work with regards to the application of control systems theory to human temperature regulation suggests skin temperature variable as a feedforward auxiliary mechanism acting in advance to prevent changes in core temperature (that is regulated by a feedback mechanism). This piece of reference should be added.

Werner, J. (2010). System properties, feedback control and effector coordination of human temperature regulation. European journal of applied physiology, 109(1), 13-25.

Author Response

Reviewer 1

Reviewer’s comments (submitted paper ID:life-1252446)

This study investigated whether the feelings of thermal comfort and sensation during exposure to alternating hot and cold environments (that they could not influence; first set of experiments) coincides with the preference of the participants (in another set of experiments) to reverse the direction of the temperature change in the water perfused suits when participants thought that it was too hot or cool. A strong correlation between the upper and lower boundaries of the thermal comfort zone observed in both experiments indicated that human participants align the action of regulating thermal comfort with its perception. Furthermore, the authors demonstrated that the direction of change in the water temperature flowing inside a water-perfused suit is another determinant factor of thermal comfort and sensation added to the absolute temperature of the water. The authors should be commended for completing the experiments of an insightful study featuring an elegant experimental design. The reviewer’s specific comments are attached below.

Thank you for your comments.

L63-64: Sleep deprivation should probably be mentioned

Thank you for this suggestion. We have included the sleep deprivation in this list and cited a review by Keramidas and Botonis (2021).

L121 and L128: ‘two experimental trails’ and ‘two separate studies were conducted’. These statements are confusing, though I assume authors intend to mean the same.

Yes, you are correct. The text has been revised accordingly.

L143: ‘heating and cooling protocol was repeated thrice’. Details are needed. How was it performed? Was the order ‘Going upwards, downwards and upwards’ or any other possible combination? Was it in three separate times?

We have revised the text. It now reads:

The temperature of the water perfusing the WPS (Twps) was initially regulated at a baseline temperature of 27°C for 10 minutes. Thereafter, subjects were exposed to a heating and cooling protocol that was repeated thrice in sequence, during which Twps varied from 27°C to 42°C and back, at a rate of 1.2°C.min-1. Prior to the onset of the trials, subjects were informed of the nature of the thermal stimuli that would be administered, but were not given any verbal feedback during the trial. 

L144: How was the rate of change in the wps temperature as well as the upper and lower boundary for the water temperature selected? To my understanding, it seems that a fastest rate of wps water temperature increase during the heating phase would shift the thresholds of sensation to a lower level and the opposite would stand for the cooling phase.

Thank you for this question. The rate of change in temperature is, of course, critical. Any alterations in the rate of change during the heating and coolong phases would result in a hysteresis, and would suggest that the observed results are a function of the simulus, and not the physiological response. The system we used to regulate the temperature of the water perfusing the suit did not have a »rate control«. The observed rate of change during the heat and cooling phases was an inherent property of the system, and was constant throughout all trials.

L164: In addition to the absolute temperature of the water flowing in and out of the suit, were participants further naïve regarding the direction of change (at any given moment) in the WPS temperature, during the heating and cooling phases?

As mentioned earlier, subjects were not provided with cues during the trial. The change in the direction of the temperature simulus (i.e., heating and cooling) was perceived by all subjects, without the need for any verbal explanation.

L201-203: The present reviewer and the majority of the literature are not convinced that the tympanic temperature is a valid index for core temperature. This has to be mentioned, as a limitation of the study. If the reviewer is not mistaken, tympanic temperature data were not presented in the results section. The authors are requested to modify the section accordingly.

We concur with the reviewer that infrared tymapnometry is not the best measure of core temperature, despite its widespread use in the clinical environment. However, just as in the clinical environment, our aim was to ensure that there was no change in tympanic temperature, as anticipated based on the results of pervious studies. The following statement has been added in the Limitations section:

The saw-tooth nature of the thermal stimulus presented to the subjects with the water-perfused suit was not of a magnitude that would be anticipated to affect core temperature. Thus, the use of infrared tympanometry as an index of core temperature may certainly be considered a limitation of the study, but the measurement was used only to verify that core temperature was stable throughout the trials. Infrared tympanometry was used to ensure no relative change in core temperature, and was not used as a measure of the absolute temperature of the core region.

L205: Data should be checked whether are normally distributed and respective results should be reported.

The subjective ratings are ordinal data and therefore non-normal by definition. We have added a statement in the analysis section.

Subjective ratings are non-normally distributed, and thus the differences between the medians of the ratings of thermal perception and thermal comfort were analysed using the non-parametric Kruskal-Wallis analysis of ranks.

L217-219: This statement should be modified. The argument would rather be that the sex-related differences regarding thermal comfort and sensation were not explored in the current manuscript due to the small sample size involved. The mix of gender with such small size, in a topic were sex effect could be present (thermal sensation) is considered a weak point of the study and should be mentioned in the discussion section.

We have revised the text as suggested, and have added the following comment in the Limitations section:

Finally, the current study included male and female subjects, but statistical comparisons of thermal perception were not conducted due to the small sample size. This is an important issue and should be further explored with studies designed specifically to investigate sex-related differences in the perception of thermal comfort.

L260-264: This part of the text seems better suited for the methods rather than the results section.

We concur with the reviewer that this introductory passage is describing the method of thermal comfort regulation, but we would like to retain this passage as it reminds the reader of the manner in which the results were obtained. There is a suitable explanation of the thermal comfort regulation in the Methods section (2.1.2). 

L305-309: Results repeated in the discussion section. Core temperature values should be provided in the results section

The results presented/repeated in this section of the Discussion provide calrification of the statement regarding the peception of thermal comfort. Omitting the mention of the results will only require the reader to refer to the Results section. The section is written in this manner for better understanding.

L349: There are no blue and red dots as the picture is given in gray scale; the caption should be modified accordingly.

Thank you for alerting us to this. The original graphs were submitted in colour, and have been changed by the journal. We have revised the text.

L396-404: Reviewer agrees with the authors that the hysteresis in the perception of thermal comfort as a function of the skin temperature acts as a feed-forward mechanism meant to activate behavioural actions before the metabolically costly autonomic regulation comes into play. Werner’s work with regards to the application of control systems theory to human temperature regulation suggests skin temperature variable as a feedforward auxiliary mechanism acting in advance to prevent changes in core temperature (that is regulated by a feedback mechanism). This piece of reference should be added.

Werner, J. (2010). System properties, feedback control and effector coordination of human temperature regulation. European journal of applied physiology109(1), 13-25.

Thank you for reminding us of this seminal work by Prof, Jurgen Werner.

Reviewer 2 Report

This is a well designed experimental study on temperature perception in humans. Also, behavioural responses to body heating and cooling were measured.

In the Introduction the pertinent literature is summarized to emphasize the need to fill the gap in our knowledge concerning the details and efficinency of physiological and behavioural regulatory means to prevent excessive alterations of body temperature that may endanger balanced bodily functioning. The experimental design aiming at modification of peripheral thermal input is described in sufficient detail for the reader to follow and understand the basis of the experimental sessions carried out. The main results obtained are depicted by clearly formulated figures and tables. The Discussion puts the main findings of these experiment in perspective of the pertinent literature. The authors mention some limitations in connection to the methods used for temperature measurements. A (too) short Summary closes the paper.

Some questions and suggestionstions

  1. From the last two sentences of the Abstract it is not clear whether the authors consider a real connection between ratings of thermal confort and behavioural actions to achieve thermal neutrality. I would suggest to add „in earlier studies” after ”for humans” (line 28). In addition, I would put „it is” after „Although” at the beginning of the same sentence.
  2. In line 30, instead of „for considering” I  woud write „to consider”.
  3. To support the idea of hysteresis mentiond in the manuscript, I would design and use a figure to demonstrate it.
  4. What was the reason to measure core temperetaure of the subjects, if these data were not even mentioned in the Results and/or in the Discussions? Alternatively, it would have been interesting to show the behaviour of core temperature during heating or cooling and compare it to the changes of skin temperature developing simultaneously.
  5. It would be helpful for the reader of the manuscript to list the abbreviatios used in text either at the beginning or end of the paper.

Author Response

Reviewer 2

This is a well designed experimental study on temperature perception in humans. Also, behavioural responses to body heating and cooling were measured.

In the Introduction the pertinent literature is summarized to emphasize the need to fill the gap in our knowledge concerning the details and efficinency of physiological and behavioural regulatory means to prevent excessive alterations of body temperature that may endanger balanced bodily functioning. The experimental design aiming at modification of peripheral thermal input is described in sufficient detail for the reader to follow and understand the basis of the experimental sessions carried out. The main results obtained are depicted by clearly formulated figures and tables. The Discussion puts the main findings of these experiment in perspective of the pertinent literature. The authors mention some limitations in connection to the methods used for temperature measurements. A (too) short Summary closes the paper.

Some questions and suggestionstions

  1. From the last two sentences of the Abstract it is not clear whether the authors consider a real connection between ratings of thermal confort and behavioural actions to achieve thermal neutrality. I would suggest to add „in earlier studies” after ”for humans” (line 28). In addition, I would put „it is” after „Although” at the beginning of the same sentence.

Thank you for this comment. We have revised the text accordingly.

  1. In line 30, instead of „for considering” I  woud write „to consider”.

Corrected.

  1. To support the idea of hysteresis mentiond in the manuscript, I would design and use a figure to demonstrate it.

As suggested, Figure 7 (caption below) has been added to the manuscript to illustrate the hysteresis:

Figure 7. Illustration of the hypothesized hysteresis responses of the perception of thermal comfort (left panel) and temperature (right panel) to heating (grey arrows) and cooling (black arrows) stimuli provided by the water perfused suit (WPS). The perception of thermal comfort (left panel, rating =0) and thermoneutrality (right panel, rating =0) occurred at higher temperatures of the water perfusing the WPS (Twps) during cooling (black lines and arrows) compared to heating (dashed lines and grey arrows).

  1. What was the reason to measure core temperetaure of the subjects, if these data were not even mentioned in the Results and/or in the Discussions? Alternatively, it would have been interesting to show the behaviour of core temperature during heating or cooling and compare it to the changes of skin temperature developing simultaneously.

Infrared tymapnometry is not the best measure of core temperature, despite its widespread use in the clinical environment. However, just as in the clinical environment, our aim was to ensure that there was no change in tympanic temperature, as anticipated based on the results of pervious studies. The following statement has been added in the Limitations section:

The saw-tooth nature of the thermal stimulus presented to the subjects with the water-perfused suit was not of a magnitude that would be anticipated to affect core temperature. Thus, the use of infrared tympanometry as an index of core temperature may certainly be considered a limitation of the study, but the measurement was used only to verify that core temperature was stable throughout the trials. Infrared tympanometry was used to ensure no relative change in core temperature, and was not used as a measure of the absolute temperature of the core region.

  1. It would be helpful for the reader of the manuscript to list the abbreviatios used in text either at the beginning or end of the paper.

Should the editor allow this, we will add it. We have not come across a list of abreviations in articles published by this journal.

Reviewer 3 Report

This work presents a progression of previous work regarding the thermal comfort zone by some of the authors, examining shift in the TCZ with varied directions of temperature change (warming, cooling). The manuscript is generally well organized and easy to follow. The authors may want to consider labelling the two protocols as 1 and 2 (or similar) for ease of reading. See specific comments below.

Specific Comments

Please emphasize how this work builds on prior studies (particularly by some of the authors) and its applications.

Page 3: Table 1. Individual data does not necessarily need included (unless specified by the journal). Consider Mean +/- SD for each sex and overall.

Page 3, Section 2.1: Please indicate if the two trials were randomized or counterbalanced.

Page 4: Nice inclusion of a schematic.

Page 4, section 2.1.1: Can the authors explain/justify the selection of the specific 1.2C ramping protocol and selection of a 3C change to obtain thermal perception/discomfort data. In other studies (e.g. Ciuha et al,. Physiol 528 Behav 2019, 210: 112623.) a 1C/min increase or decrease in Ta was utilized and thermal comfort questions to determine the TCZ were asked every 3 min (versus 3C in the present protocol).

Page 4, section 2.1.1: Is there a reference for the scales (e.g. ISO?).

Page 4, Line 155: Should the temperature change be 1.2C per min (versus the 1C indicated)? It states the protocol is the same as the ‘perception of thermal comfort trial’. Correct or justify the difference in experimental protocol.

Page 4, sections 2.1.1. and 2.1.2: Were baseline measures recorded at the end of the familiarization period at 25c/30%? It would be useful to know the starting point of all subjects.

Page 5, section 2.2.3: Were the ambient conditions maintained (Ta/RH) or simply measured? Reference was made to specific conditions for the pre-test period.

Page 6, Lines 230-231: No significant differences in median votes between trails? Please clarify.

Page 6 and 7, Figures 2/3: In the legend, include that median values are presented. It would be useful to include the interquartile ranges on the graphs (although I appreciate you have 3 trails overlapped).

Page 6, Table 2: The TCZ data are presented as mean +/SD but other data are presented as medians. Please justify.

Page 11, section 5: Consider revising ‘demonstrates’ to ‘suggests’ or something similar regarding the feedforward mechanisms. It may be a little strong to indicate this study demonstrates feedforward when a considerable portion is speculative (albeit with rationale explanations proposed).

Do the authors believe the methodological approach (e.g. WPS) impacts the thermal comfort versus circumstances that allow evaporation? This may be worth further discussion.

Considering this manuscript and other work by this group, the TCZ varies based on the direction and rate of temperature change. Can the authors please add a brief section regarding the importance and applications of this work. This may also relate to the prior comment.

Author Response

This work presents a progression of previous work regarding the thermal comfort zone by some of the authors, examining shift in the TCZ with varied directions of temperature change (warming, cooling). The manuscript is generally well organized and easy to follow. The authors may want to consider labelling the two protocols as 1 and 2 (or similar) for ease of reading. See specific comments below.

Specific Comments

Please emphasize how this work builds on prior studies (particularly by some of the authors) and its applications.

This has been added in the Conclusions.

Page 3: Table 1. Individual data does not necessarily need included (unless specified by the journal). Consider Mean +/- SD for each sex and overall.

Table deleted, and Mean ± SD information added to text.

Page 3, Section 2.1: Please indicate if the two trials were randomized or counterbalanced.

The order of the two trials were randomised. This has been added to the text.

Page 4: Nice inclusion of a schematic.

Thank you.

Page 4, section 2.1.1: Can the authors explain/justify the selection of the specific 1.2C ramping protocol and selection of a 3C change to obtain thermal perception/discomfort data. In other studies (e.g. Ciuha et al,. Physiol 528 Behav 2019, 210: 112623.) a 1C/min increase or decrease in Ta was utilized and thermal comfort questions to determine the TCZ were asked every 3 min (versus 3C in the present protocol).

The suit and control system are of our design. The control systems for heating and cooling the water were different. We did not design the control system in such a manner that we could vary the rate of cooling and heating. The rates were determined by the characteristics of the control systems. The system used in the study by Ciuha et al. maintained a rate of heating and cooling of 1°C, wheras the system used in the present study maintained a heating/cooling rate of 1.2°C

Page 4, section 2.1.1: Is there a reference for the scales (e.g. ISO?).

This is a standard questionnaire, which is used in thermoregulatory research. We have provided the descriptors of the visual analog scale in the text.

Page 4, Line 155: Should the temperature change be 1.2C per min (versus the 1C indicated)? It states the protocol is the same as the ‘perception of thermal comfort trial’. Correct or justify the difference in experimental protocol.

Corrected.

Page 4, sections 2.1.1. and 2.1.2: Were baseline measures recorded at the end of the familiarization period at 25c/30%? It would be useful to know the starting point of all subjects.

No measurements were made during the 30 min period of acclimation to the temperature and relative humidity within the laboratory.

Page 5, section 2.2.3: Were the ambient conditions maintained (Ta/RH) or simply measured? Reference was made to specific conditions for the pre-test period.

The trials were not conducted in a climatic chamber, but in a termperature controlled room (i.e. air conditioning and radiators). The temeprature and humidity of the ambiet air within the labroatory was measured, but not actively maintained. Since the trials were conducted within the same period of the year, the temperature and relative humidity was stable during the course of the study.

Page 6, Lines 230-231: No significant differences in median votes between trails? Please clarify.

Should read:

There were no significant differences between the median votes obtained from the three trials over the given range of temperatures.

Text has been corrected.

Page 6 and 7, Figures 2/3: In the legend, include that median values are presented. It would be useful to include the interquartile ranges on the graphs (although I appreciate you have 3 trails overlapped).

Comment that values are median included in legend of figure. Including the interquartile ranges would make the graphical presentation unnecessarily complex/busy. We would prefer to leave the graph as it is.

Page 6, Table 2: The TCZ data are presented as mean +/SD but other data are presented as medians. Please justify.

The TCZ data is ratio data, whereas the subjective ratings ar ordinal data. Ratio data may be  represented as averages with standard deviations, whereas ordinal data is presented as medians.  

Page 11, section 5: Consider revising ‘demonstrates’ to ‘suggests’ or something similar regarding the feedforward mechanisms. It may be a little strong to indicate this study demonstrates feedforward when a considerable portion is speculative (albeit with rationale explanations proposed).

Corrected.

Do the authors believe the methodological approach (e.g. WPS) impacts the thermal comfort versus circumstances that allow evaporation? This may be worth further discussion.

Thank you for this question. We certainly believe that our model might explain the perception of thermal comfort during evaporation of sweat from the skin. This, however, is a speculation, which we did not wish to include in the text. It is certainly an issue which warrants further experimental validation.

Considering this manuscript and other work by this group, the TCZ varies based on the direction and rate of temperature change. Can the authors please add a brief section regarding the importance and applications of this work. This may also relate to the prior comment.

Thank you for this comment. We have added comments regarding our previous work in different sections of the manuscript.